# Developing a Simple, Effective, and Quick Process to Make Silver Nanowires with a High Aspect Ratio

**DOI:** 10.3390/ma16155501

**Published:** 2023-08-07

**Authors:** Gharam A. Alharshan, Mohamed A. M. Uosif, Rabeea D. Abdel-Rahim, El Sayed Yousef, Essam Ramadan Shaaban, Adham M. Nagiub

**Affiliations:** 1Physics Department, College of Science, Princess Nourah Bint Abdulrahman University, P.O. Box 84428, Riyadh 11671, Saudi Arabia; 2Physics Department, College of Science, Jouf University, Sakaka P.O. Box 2014, Saudi Arabia; 3Chemistry Department, Faculty of Science, Al-Azhar University, Assuit 71524, Egypt; 4Research Center for Advanced Materials Science (RCAMS), King Khalid University, P.O. Box 9004, Abha 61413, Saudi Arabia; 5Physics Department, Faculty of Science, King Khalid University, P.O. Box 9004, Abha 61413, Saudi Arabia; 6Physics Department, Faculty of Science, Al-Azhar University, Assiut 71542, Egypt

**Keywords:** silver nanowires, transparent conductive electrode, flexible electrodes, polyol method

## Abstract

A growing number of people are interested in using silver nanowires (AgNWs) as potential transparent and conductive materials. The production of high-performance and high-throughput AgNWs was successfully optimized in this work using a one-step, straightforward, and reproducible modified polyol approach. The factors influencing the morphology of the silver nanowires have undergone extensive research in order to determine the best-optimized approach for producing AgNWs. The best AgNW morphology, with a length of more than 50 m and a diameter of less than 35 nm (aspect ratio is higher than 1700), was discovered to be produced by a mixture of 44 mM AgNO_3_, 134 mM polyvinylpyrrolidone (PVP) (Mo.Wt 40,000), and 2.4 mM KCl at 160 °C with a stirring rate of 100 rpm. With our improved approach, the overall reaction time was cut from almost an hour with the conventional polyol method to a few minutes. Scanning electron microscopy (SEM), X-ray diffraction (XRD), and ultraviolet (UV) spectroscopy were used to characterize AgNWs. The resultant AgNWs’ dispersion was cleaned using a centrifuge multiple times before being deposited on glass and PET substrates at room temperature. In comparison to commercial, delicate, and pricey indium-doped tin oxide (ITO) substrates, the coated samples displayed exceptionally good sheet resistance of 17.05/sq and optical haze lower than 2.5%. Conclusions: Using a simple one-step modified polyol approach, we were able to produce reproducible thin sheets of AgNWs that made excellent, flexible transparent electrodes.

## 1. Introduction

One-dimensional nanomaterials have attracted the attention of scientists and researchers in recent years. Nanowires are an important type of one-dimensional nanostructure, with a diameter of fewer than 100 nanometers and a length ranging from a few hundred nanometers to tens of microns [1]. It is worth mentioning that the development of silver nanowires (AgNWs) is studied because of their unique and excellent optical, mechanical, and electromagnetic properties, allowing them to be introduced in many modern applications, such as catalysis, sensors, optoelectronic devices, and nano electronics [2]. Silver has been found to have the highest thermal and electrical conductivity among all metals compared to other metals [3,4,5]. Therefore, many attempts were conducted to obtain silver transparent conductive electrodes (TCEs) [6,7,8].

As a result of the unique optical and electrical properties of silver nanowires, there are many applications, based on AgNWs. Examples of applications based on silver nanowires are flexible displays, biosensors, electronic textiles, artificial organs, and other portable and wearable electronic devices In addition to applications in power supplies and transparent conductive electrodes. Most photovoltaic devices depend on transparent conductive electrodes to be suitable for some desired photovoltaic applications, so silver nanowires were the best for designing transparent electrodes [9,10,11,12].

Silver nanowires are synthesized by more than one method, including hard template and soft template methods [13]. Examples of the hard template method have been performed using deoxyribonucleic acid (DNA) templates [14] and nanoporous membranes template [15]. The soft template method was conducted in the past ten years by using cetyltrimethylammonium bromide (CTAB) as a capping agent [16], polyvinyl alcohol [17], double-hydrophilic block copolymers [18], and polyvinyl pyrrolidone, which is known as the polyol method [19]. In 1989. Fievet et al synthesized nanostructures for the first time through polyol technology [20]. The polyol method is the most common and the simplest method for large-scale and high-quality production of AgNWs, and it does not require much technical knowledge. However, it is not easy to produce AgNWs of uniform shape and size, and the final products are usually contaminated with silver nanoparticles (AgNPs) that must be removed from AgNWs to obtain AgNWs with good optical and electrical properties [21].

The morphology of AgNWs is extremely sensitive to the reaction environment and elements thereof, such as reaction temperature, the molar ratio of chemicals used in the synthesis, and the agitation speed [5]. It has also been reported that AgNWs produced by the polyol process depend on the type of PVP used as a capping agent [22]. Different solvents that simultaneously acted as reducing agents have been used in the preparation of AgNWs, including ethylene glycol (EG) [23], glycerol, and propylene glycol [23]. 

The previous strategies, such as the hard method and its types, are complicated, require special conditions, and take a long time. Usually, the traditional polyol technique requires more than 12 h in an environment filled with inert gas and involves multiple steps [24,25,26]. Additionally, the produced nanowires lack controlled morphology, which serves as our motivation for undertaking this research [27,28]. In the present study, we specifically concentrated on creating AgNWs using a quick and simple modified polyol approach. By improving the reaction conditions, the reaction time was significantly shortened, and AgNWs with an excellent aspect ratio of approximately 1766 were produced. Additionally, the cleaned AgNWs solution was effectively coated on glass and PET substrates, producing materials with good transparency (78%) and low electrical resistance (17.05/sq). As a result, the resulting electrodes make an excellent ITO replacement.

## 2. Materials and Methods

All the chemicals used for this work were of high analytical quality and did not require further purification. Silver nitrate (AgNO_3_), ethylene glycol (EG), glycerol, propylene glycol, KCl, KBr, and Polyvinylpyrrolidone (PVP) (Molecular Weight ≈ 400,000) were all purchased from Sigma Aldrich (St. Louis, MO, USA). 

### 2.1. Synthesis of AgNWs

First, 10 mL of ethylene glycol, 5 mL of KCl (0.005 g of KCl in 5 mL of ethylene glycol), 0.3 g of PVP solution (50% in ethylene glycol), and 0.15 g AgNO_3_ were mixed in a single container and heated to 160 °C until the desired color was obtained. The reaction was then stopped by cooling it with an ice bath [24]. Afterward, the product was washed several times with distilled water and ethanol using a centrifuge at 2500 rpm. The final product was re-suspended in ethanol.

### 2.2. Electrodes Designing

AgNWs suspension solution of 1% was well re-dispersed using ultrasonication for 10 min. The AgNWs were made into formulated AgNWs ink using 0.1% by weight of chitosan (MW~10,000) as a thickening agent, and the clean AgNWs suspension (0.2%) was then diluted to (0.05% by weight). The final AgNWs suspension (ink) was then coated by using a spin coater at different steering speeds for one minute on a clean glass substrate. To coat AgNWs on PET sheets, PET was first immersed in H_2_O_2_ for 2 h to become hydrophilic and then washed with ethanol several times before use. A Meyer rod was used to coat AgNWs on the pretreated PET sheets. 

## 3. Morphological Characterization

X-ray diffraction spectra were collected using a Philips PW 1710 (Tokyo, Japan) V-530 X-ray diffractometer, which employed Cu Kα radiation at 40.1 V and 30 mA with a wavelength of 0.154 nm. Scanning Electron Microscopy images were taken with a Joel (Tokyo, Japan) JSM 5600 LV Scanning Electron Microscope equipped with an Oxford Instruments 6587 EDX (Energy Dispersive X-Ray) (Energy Dispersive X-Ray) Microanalysis detector. Additionally, UV spectroscopy was conducted using a JASCO Model V-530 UV-Vis spectrophotometer (Tokyo, Japan).

## 4. Results and Discussion

The reduction of silver atoms using EG, which is used as a solvent and at the same time acts as a reducing agent, at an elevated temperature of about 160 °C can be represented according to the following’s chemical equations [29]:



As shown in Equation (1), in the presence of halide anions as metal chlorides or metal bromides, AgNO_3_ reacts with chlorides or bromides to form AgX, which promotes the reduction process of Ag^+^ and can control silver concentration [30,31]. In the second step, Ag^+^ ions were reduced by EG to form silver atoms (seeds) as explained in Equations (2) and (3). The concentration of the seeds reaches the level of supersaturation at which the nucleation of Ag atoms takes place, and they start to grow into silver nanostructures in the solution phase [27,28]. 

PVP works as a good capping agent, and it plays a significant role in the final morphological shape of the silver nanostructure [5,32]. PVP has a high ability to form coordination bonds with many chemical compounds. The coordination properties are due to the skeleton of PVP which has a strong polar group (pyrrolidone ring). Carbonyl polar groups (C=O) were coordinated with Ag^+^ ions to form the PVP-Ag complex, as shown in Figure 1. Ag^+^ ions were absorbed on the active sites of the PVP polymer surface, and then Ag ions were converted to nuclei of AgNWs [30]. 

The AgNWs’ nuclei were growing in one direction of the face-centered-cubic (FCC) (111) structure, which is the available face crystal growth as is proposed and confirmed with X-ray diffraction analysis in the following section. As proposed in Figure 1, PVP was covering the (100) plan and accordingly, Ag crystal was blocked from growth through the (100) plan and forced to grow through the (111) plan to form AgNWs. It has been reported that an excess amount of PVP will bind to the (111) plan of the Ag crystalline structure, which will lead to high coverage of every facet of the silver crystal nucleus, and therefore no AgNWs will be formed [27,28]. 

### 4.1. X-ray Diffraction (XRD)

The XRD pattern of the AgNWs film coated on a glass substrate (Figure 2) was analyzed and found to have a high crystalline structure and compatibility with Joint Committee on Powder Diffraction Standards (JCPDS) File No. 98-018-0878, which reflects a cubic crystalline structure [33]. Five distinct characteristic patterns at 38.37, 44.6, 64.91, 77.991, and 82.18° were observed in the XRD data, corresponding to Miller Indices in (FCC) (111), (200), (220), (311), and (222) planes, respectively. The crystal lattice structure constant was determined to be 4.0861 Å, which agrees with the standard reported value of 4.0862 Å [34,35,36].

For AgNWs, XRD examination yielded crystal lattice (d) spacing values of 2.31 and 2.14 A°, respectively, which correspond to the planes (111) and (200). These results demonstrated that samples of AgNWs were effectively synthesized into high-purity crystalline structures [37]. Additionally, the intensity ratio between the (111) and (200) peaks was shown to be 3.5 compared to the theoretical value of 2.5, which may indicate the enhancement of the (111) crystal plane in AgNWs as mentioned in the previous studies [27,28,29,38].

### 4.2. UV-Spectroscopy

Plasmon surface resonance frequencies are frequently employed in UV-visible spectroscopy to describe the morphological characterization of silver nanostructures [39,40]. The result of a conventional polyol-based AgNWs synthesis comprises silver nanowires, nanoparticles, and other metallic detritus. This combination may be separated using adequate centrifugation to obtain practically pure AgNWs. Figure 3 depicts the UV-spectroscopy of AgNWs before and after centrifuge cleaning. AgNPs peaked at 419 nm in UV absorbance peaks, which can be attributed to pre-cleaning the AgNWs (Figure 3A). To preserve additional information, Gaussian fitting was used for the collected UV peaks. Three distinct peaks were seen at 343, 375, and 419 nm (Figure 3B). The first and second peaks were assigned to the AgNWs, whereas the third peak was assigned to the AgNPs [36]. Figure 3C depicts cleaned AgNW samples, whereas Figure 3D depicts the Gaussian fitting. Only two peaks associated with AgNWs were found at 343 and 375 nm, whereas the third peak associated with AgNPs vanished, indicating that the cleaning method was successful in eliminating nanoparticles [37]. The presence of two AgNW generations with slightly varying diameters may explain the development of two AgNW peaks [38]. The findings reveal that the location of the AgNW surface plasmon resonance peak is highly dependent on wire diameter.

### 4.3. Polyol Method Versus One-Step Modified Polyol Method

Silver nanowires were synthesized by the reduction of AgNO_3_ by ethyelen glycol in the presence of PVP [3,5,41,42]. A comparison was made between the traditional polyol method, which requires several steps, and our one-step modified polyol method. All chemicals used, their temperatures, and the agitation speed were kept the same in the two methods.

Figure 4 shows SEM images and their corresponding histogram distribution for AgNWs prepared by the traditional polyol method (Figure 4A–D) and the new modified polyol method (Figure 4E–H). Different SEM magnifications of 10 and 500 nm for the same samples were taken and were used for calculating the average AgNWs lengths and diameters shown in the corresponding histograms. The average length of AgNWs synthesized by the traditional polyol method was found to be 18.47 μm while their diameters were about 50 nm. On the other hand, the average length and diameter for AgNWs obtained by one-step modified polyol were found to be 58.25 μm and 34.33 nm, respectively. The aspect ratio for the traditional polyol method was 369.4 while for the modified method was 1664.2. SEM images also showed that the new method has a more uniform and clean wire distribution than the traditional method at the same reaction conditions. 

Based on the previous data, AgNWs produced by the new modified polyol method had a better aspect ratio and highly pure and uniform AgNWs compared to the traditional polyol method. Moreover, the reaction time was reduced from 45 min to about 8 min. 

### 4.4. Parameters Affecting the Synthesis of AgNWs

AgNWs’ crystal growth is strongly affected by both chemical and physical conditions during its synthesis. In this current work, we optimized the methodology of AgNWs synthesis by studying the main parameters that affect the synthesis such as PVP: AgNO_3_ molar ratio, temperature, agitation speed, solvent, and halide salt type.

#### AgNO_3_ Concentrations

It has been reported that the morphology of Ag nanostructures obtained by the polyol method is strongly influenced by the silver source concentration [3,43,44]. Figure 5 shows SEM images of AgNWs synthesized by four different AgNO_3_ concentrations. A very thin wire with an average diameter of 25 nm and an average length of 8.49 μm. Additionally, a lot of nanoparticles were produced with 20.6 mM AgNO_3_ (Figure 5A–C). Using a higher concentration of AgNO_3_ produces thicker AgNWs without AgNPs (Figure 5D–L). Average diameters of 49.87, 137, and 270 nm and average lengths of 25.87, 30.37, and 39.17 μm were observed for 44, 88.3, and 176.6 mM of AgNO_3_, respectively.

Figure 6A shows the effect of AgNO_3_ concentration on the average diameter of AgNWs. The average diameter was increased by increasing AgNO_3_ concentration, whereas the average AgNWs length reached a maximum at an AgNO_3_ concentration of 44 mM (Figure 6B).

The UV-absorbance spectra of AgNWs synthesized with different AgNO_3_ concentrations are shown in Figure 7. Five characteristic UV absorbance peaks were observed at 368, 376, 390, 397, and 410 nm which correspond to AgNWs synthesized with four different AgNO_3_ concentrations. The first three peaks were attributed to AgNWs with average diameters of 270, 137, and 49.87 nm which were prepared under 176.6, 88.3, and 44 mM of AgNO_3_ concentrations, respectively. The two other peaks of the characterized AgNWs had an average diameter of 25.43 nm. Based on these data, with increasing the concentration of silver nitrate, the thickness of the wires increases, which leads to the emergence of UV absorption values towards a redshift (Figure 7A) [45,46]. Figure 7B,C show the UV absorbance spectra for the AgNW suspension sensitized with 20.6 mM AgNO_3_ and its corresponding Gaussian fitting, respectively. Two UV absorbance peaks were observed at 368 and 418 nm, which were assigned to the thinner AgNWs and AgNPs, respectively [40]. Very thin AgNWs can be obtained by using a low concentration of AgNO_3_; however, silver nanoparticles will also be obtained as indicated by the presence of a UV peak at 418 nm. It may be concluded that using a low concentration of silver source not only will yield AgNWs with thinner diameters, but the suspension will also contain silver nanoparticles. Consequently, to obtain uniform and thin wire with a minimum amount of AgNPs, a concentration of 44 mM was applied in this study. 

### 4.5. Agitation Speed

The agitation speed of chemical reactions highly affects the growth of AgNWs crystals and should be carefully adjusted [47]. Figure 8 shows SEM images of AgNWs synthesized with different agitation speeds ranging from 0 to 700 rpm. As was expected, different agitation speeds will lead to different AgNW morphologies. The average diameters of AgNWs obtained with agitation speeds of 700, 300, 100, and 0 rpm were found to be 23.866, 42.67, 66.5, and 104.5 nm, while the lengths of the wires were 16.08, 41.066, 56.86, 54.506 μm, respectively. Additionally, SEM images proved that AgNWs synthesized with controlled agitation will produce more uniform AgNWs than those obtained without agitation (Figure 8L). 

Figure 9 summarizes the dependence of AgNW morphology on agitation speed. Figure 9A shows that the average length of the synthesized AgNWs slightly increased at 125 rpm and then gradually decreased with increasing agitation speed. On the other hand, Figure 9B shows that AgNWs prepared with high agitation speeds were thinner than those obtained at low agitation speeds. These results agree with previous reports [47,48]. This behavior may be attributed to the fact that the increased agitation speed leads to high surface energy on the face of (111) that facilitates the Ag seeds to form multi-twin particles, and accordingly, small seeds are produced, which leads to the formation of a thinner wire [47]. 

### 4.6. Temperature

Temperature is believed to be significant in controlling the growth of AgNWs crystals [49]. Figure 10A–E demonstrate SEM images of AgNWs obtained at different temperatures at 110, 135, 160, 175, and 190 °C, while Figure 10F shows the relation between temperature and average AgNW length. The length of AgNWs is affected by the reaction temperature, as indicated by the gradual increase in the average length of AgNWs at a temperature range between 110 and 160 °C, followed by a rapid decrease at a temperature above 160 °C, until 195 °C. This decrease in average wire length may be attributed to the destruction of AgNWs being produced at temperatures over 175 °C.

Figure 11A–E illustrate SEM images of AgNWs obtained at different reaction temperatures. The relation between average AgNWs diameters and different reaction temperatures is shown in (Figure 11F). A decrease in average AgNWs diameter with an increase in the synthesis temperature until 160 °C followed by a slight increase were observed at a temperature higher than 175 °C. It might be concluded that by increasing the reaction temperature, AgNWs will be longer and thinner in agreement with previous reports [50]. A temperature of 160 °C was found to be the optimum temperature condition for AgNWs in this study. 

As was mentioned earlier, reducing reaction time is of great economic significance. In this study, the time required to form the wire at 110 °C was about 12 h. Figure 12 shows the relation between reaction temperature and reaction time. The time required to produce AgNWs decreased from 12 h to 20 min when the reaction temperature was raised from 110 to 160 °C. These results are consistent with previous research [32,49]. As was found in the literature, using temperatures between 160 to 180 °C will produce good AgNWs, but from an economic point of view, using elevated temperatures for a longer time will increase the cost of AgNW production. Therefore, reducing the reaction time in our one-step synthesis to 8 min compared to 45 min in the traditional polyol method is an advantage.

### 4.7. Halide Type

The addition of metal halide ions was found to facilitate the formation of AgNW crystal growth [51,52,53,54,55]. In the current study, Br^−^, Cl^−^, and I^−^. were used as co-nucleants to obtain AgNWs. Figure 13 shows SEM images of AgNWs synthesized by using different Potassium halides. Ultrathin AgNWs (20 nm) were obtained by using 0.005 g KBr which functioned as a co-nucleant that may inhibit the lateral growth of nanowires Figure 13A–C. An average length of 23.6 μm and an average diameter of 21.67 nm were obtained when using Br as a source of halogen in this study (Figure 13C). When using chloride ions, an average length of 58.25 μm and an average diameter of 34.3 nm were obtained (Figure 13D–F). On the other hand, Ag debris, AgNPs, and very few wires were observed when using iodide salts as shown in (Figure 13G–I). 

Accordingly, it can be concluded that AgNWs obtained with bromide ions will be thinner and shorter than those obtained with chloride ions. Few wires were obtained when using an iodide source at the same reaction conditions. AgNWs produced with a chloride source will have a much better aspect ratio of about 1700 compared to 1089 obtained with a bromide source. In agreement with previous research, thinner wires were produced with bromide ions [56] and longer with chloride ions; [57] however, iodide is not valid for preparing AgNWs. 

Figure 14 shows SEM images for AgNWs synthesized with different concentrations of potassium chloride. By increasing the amount of KCl, AgNWs not only become thinner but also shorter and more nanoparticles and debris will be obtained. Figure 15A,B summarizes the effects of potassium chloride concentration on the AgNWs’ morphologies. AgNWs’ average length reached a maximum length of about 55 microns at a KCl concentration of 2.6 mM, whereas the average length gradually decreased with the increase in KCl concentrations (Figure 14A). The diameters of AgNWs became thinner by increasing potassium chloride concentration, which is in agreement with the previous studies (Figure 15B) [57]. 

Figure 15C shows the UV absorbance peaks for AgNWs synthesized with different KCl concentrations. Four UV peaks at 401, 395, 387, and 364 nm were observed, corresponding to KCl concentrations of 0.86, 2.6, 4.4, and 8.94 mM, respectively. In consistency with previous results, it was found that by increasing the amount of halogen, the diameter of the resulting wire was reduced as indicated by the appearance of a UV peak of a shorter wavelength [45]. Based on the data shown in Figure 14, using an optimum concentration of 2.6 mM KCl will produce AgNWs with a good aspect ratio (Figure 14B). Other KCl concentrations will show nanoparticles and debris with shorter AgNWs. 

### 4.8. AgNWs Transparent Conductive Electrode (AgNWs-TCE)

High-performance transparent conductive electrodes should have excellent optical clarity, excellent electrical conductivity, good mechanical properties, and flexibility. These characteristics can be achieved by increasing the aspect ratio of the silver nanowires. A spin coater was used to fabricate AgNWs electrodes on a glass substrate followed by annealing at 120 °C for 5 min to design a conductive transparent electrode. Different transparent conductive electrodes were designed by coating with different spin coating rates of 500 rpm, 1000 rpm, 2000 rpm, and 3000 rpm for 1 min.

Figure 16 shows both the optical and electrical properties of AgNWs-TCE prepared at different spin coating rates. Transmittance of AgNWs electrodes prepared at spin frequencies of 500, 1000, 2000, 3000, and 4000 rpm. The light transmittance obtained for the corresponding samples were 72.23, 77.9, 81.3, 82.6, and 91.8%, as shown in Figure 16A.

Optical haze is a basic but under-researched characteristic of transparent electrodes in solar cells, and it is also affected by the shape of the nanostructures and the thin layer thickness that makes up the electrodes. The haze factor was calculated according to the following equation: (4)$$\mathrm{Haze}\%=\frac{\;T4}{T2}-\frac{T3}{T1}$$
where *T*1 is the transmittance of the incident light, *T*2 is the total light transmitted by the sample, *T*3 is the light scattered by the instrument, and *T*4 is the light scattered by the instrument and the sample [29]. 

Figure 16B shows the optical haze of the corresponding samples. Higher spin-coating speed will result in smaller film thickness, and an accordingly lower haze and higher transmittance values are expected.

Figure 16C shows the effect of the spin coating rate on the sheet resistance of AgNWs-TCF. The sheet resistance of the AgNWs-TCF was found to be 5.5, 17.05, 27.41, 57.41, and 105.3 Ω/sq for spin coating rates 500, 1000, 2000, and 3000 rpm, respectively. The amount of AgNWs coated in glass substrate is inversely proportional to the spin rate. Therefore, increasing the amount of coating will reduce the optical transmittance and the electrical resistance [57]. The diameter and length of the nanowires are critical for transparency, haze, and surface roughness. Moreover, Figure 16D describes the relationship between sheet resistance and transmittance of AgNWs film on the glass substrates. Different images of AgNWs-TCE and their corresponding electrical resistances are shown in Figure 16E. 

Figure 17A shows a clear and uniform AgNW film coated on a glass substrate with a sheet resistance of 16.7 ohms/sq, whereas Figure 17B shows a uniform layer of flexible AgNWs on a PET substrate. The conductivity and flexibility of the designed electrode film were shown clearly through the electrical circle [58]. Figure 17C shows the AgNW ink used in the coating process. 

## 5. Comparison between the AgNW TCE with the Other Electrodes

The performance of AgNW mesh transparent electrodes was compared to indium tin oxide (ITO) and carbon nanotube (CNT) electrodes. It was found that AgNW electrodes had comparable sheet resistance and optical transmittance to ITO and CNT electrodes [59]. Additionally, AgNWs electrodes showed superior flexibility and mechanical stability compared to ITO and CNT electrodes. Another study investigated the scalability of AgNWs transparent electrodes by studying their coating properties on large-area substrates. The performance of AgNWs electrodes was compared to ITO films in terms of sheet resistance, optical transmittance, flexibility, and reliability under bending conditions. The results showed that AgNWs electrodes could achieve similar or better performance than ITO films while being more scalable and cost-effective [60]. In addition, a comparison of transparent AgNW electrodes with other works showed that the electrode resistivity of AgNWs was 27.41 Ω/sq, which was lower than the resistivity of 79 Ω/sq at a similar transparency for other works [61]. These results suggest that transparent AgNW electrodes are an excellent choice for transparent electrodes. 

## 6. Conclusions

AgNWs with an excellent aspect ratio were produced using a one-step modified polyol process. A comparison between the traditional polyol method and the one-step method was made. It was found that AgNWs produced by the improved one-step polyol method are longer and thinner than the wires obtained by the old traditional method under the same conditions. A detailed study was performed to investigate the effect of different parameters controlling the growth of silver crystals. Reaction temperature, silver source concentration, polymer type, agitation speed, and halide source are believed to be the main factors that control the growth of AgNWs. The optimum condition for our facile one-step modified polyol process was achieved. It was found that by mixing all the reactants in one pot and using a small agitation of 100 rpm, a temperature of 160 °C, 150 mM AgNO_3_, and 2.6 mM of KCl will yield AgNWs with an aspect ratio of 1764 that can be incorporated in the manufacture of optical devices. By producing a high aspect ratio of AgNWs, highly conductive transparent electrodes with fewer junctions and lower junction resistance can be fabricated. In the current study, we prepared transparent conductive electrodes at room temperature and without any post-processing procedures such as in the case of ITO electrodes. These electrodes can be made by coating on glass substrates or a flexible PET sheet. We constructed different AgNW electrodes that have low sheet resistances between 17.05 and 105.3 Ω. sq^−1^ and with transmittances greater than 76.8; 91.8% of these can be introduced in optical devices and other applications. This method may be valid for the preparation of other metal nanowires such as Cu and Al, but it requires the replacement of the capping agent with another type.

## Figures and Tables

**Figure 1 materials-16-05501-f001:**
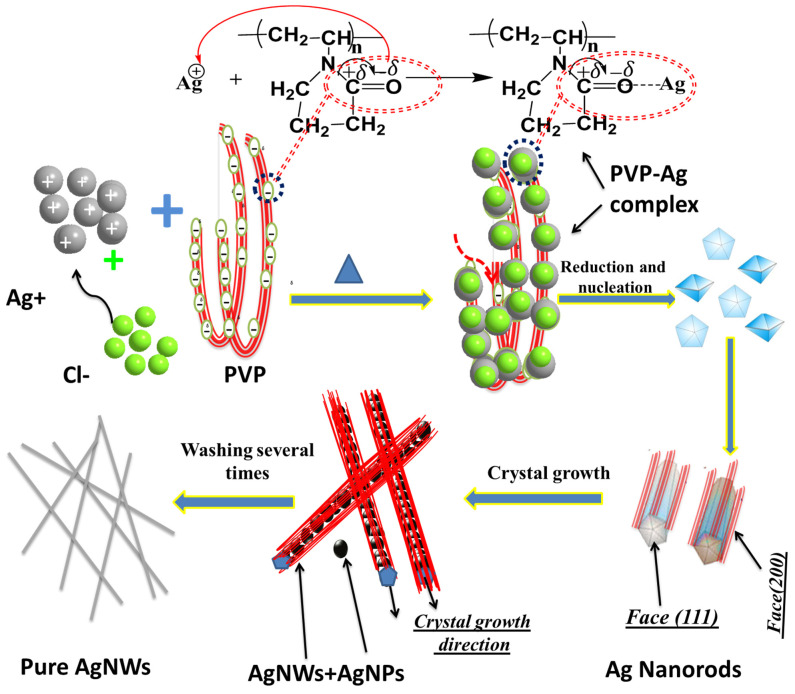
Growth mechanism of AgNWs.

**Figure 2 materials-16-05501-f002:**
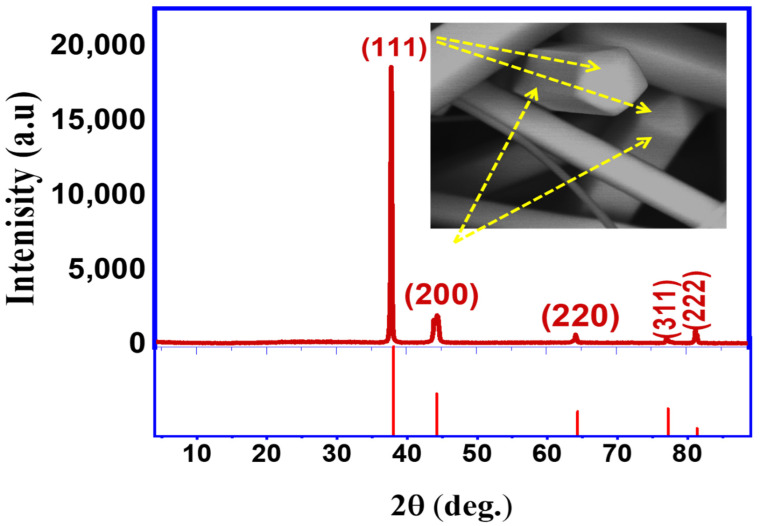
The XRD spectrum with the reference pattern of AgNWs synthesized by the one-step polyol method and its corresponding SEM image.

**Figure 3 materials-16-05501-f003:**
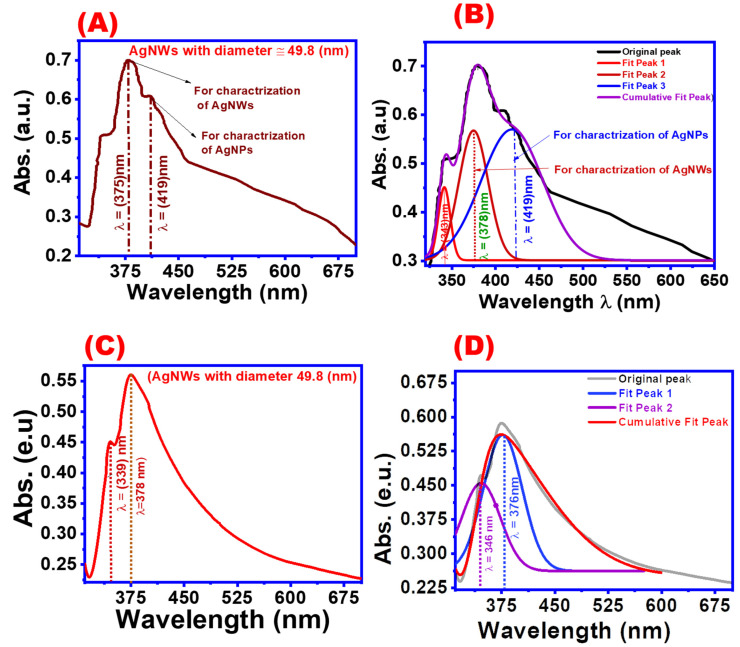
UV-visible spectroscopy of AgNW suspensions: (**A**) before cleaning, (**B**) the corresponding Gaussian fitting of UV-visible spectroscopy of silver nanowires suspension before cleaning, (**C**) after cleaning, and (**D**) the corresponding Gaussian fitting of UV-visible spectroscopy of the AgNW suspensions.

**Figure 4 materials-16-05501-f004:**
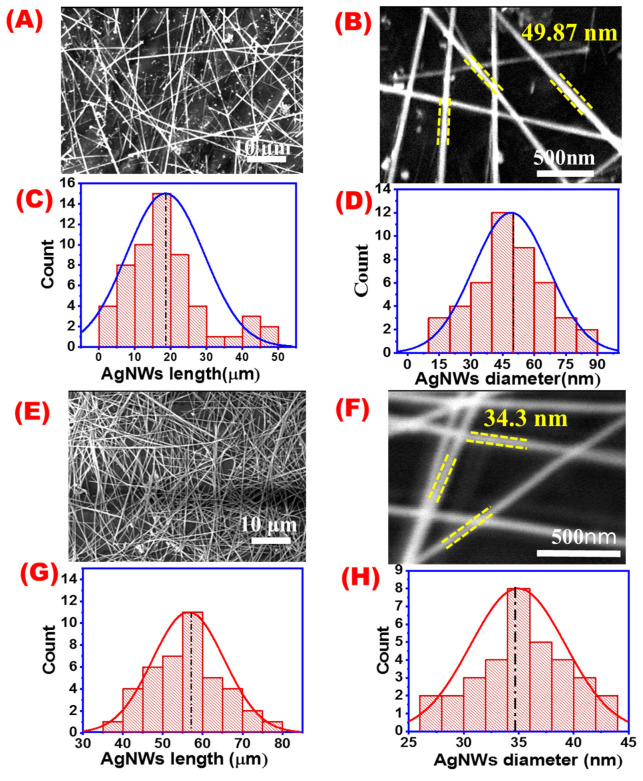
SEM images at different magnifications and their corresponding length and diameter distribution (**A**–**D**) for AgNWs prepared by the traditional polyol method; and (**E**–**H**) for AgNWs prepared by the one-step method.

**Figure 5 materials-16-05501-f005:**
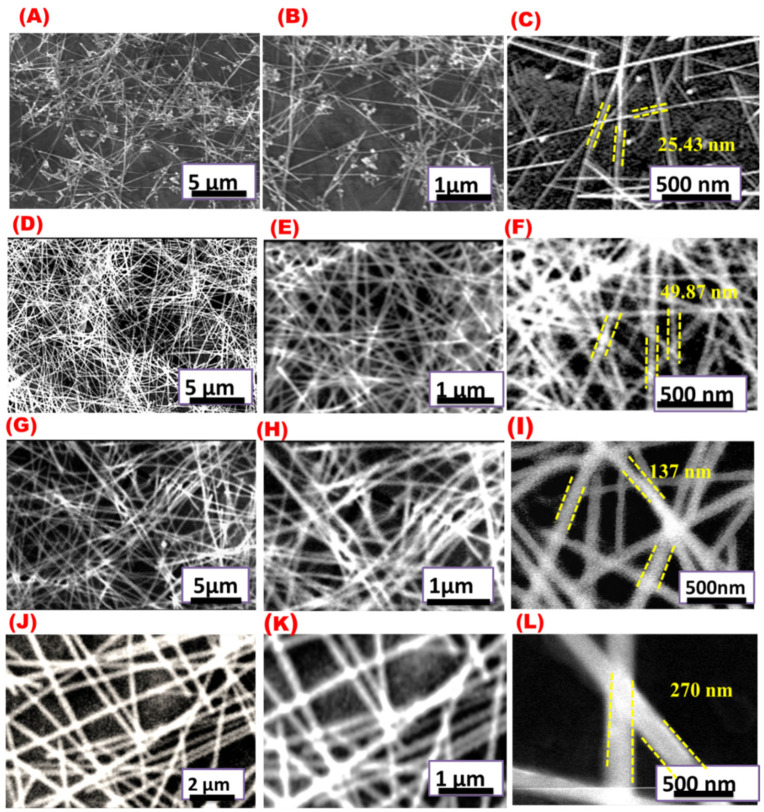
SEM images at different magnifications for AgNWs prepared with different AgNO_3_ concentrations: (**A**–**C**) 20.6 mM of AgNO_3_, (**D**–**F**) 44 mM of AgNO_3_, (**G**–**I**) 88.3 mM of AgNO_3_, and (**J**–**L**) 176.6 mM of AgNO_3_.

**Figure 6 materials-16-05501-f006:**
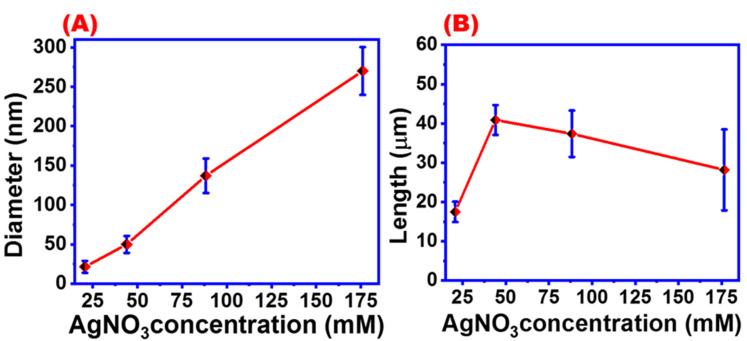
Effect of AgNO_3_ concentration on the (**A**) diameter of AgNWs and (**B**) length of AgNWs.

**Figure 7 materials-16-05501-f007:**
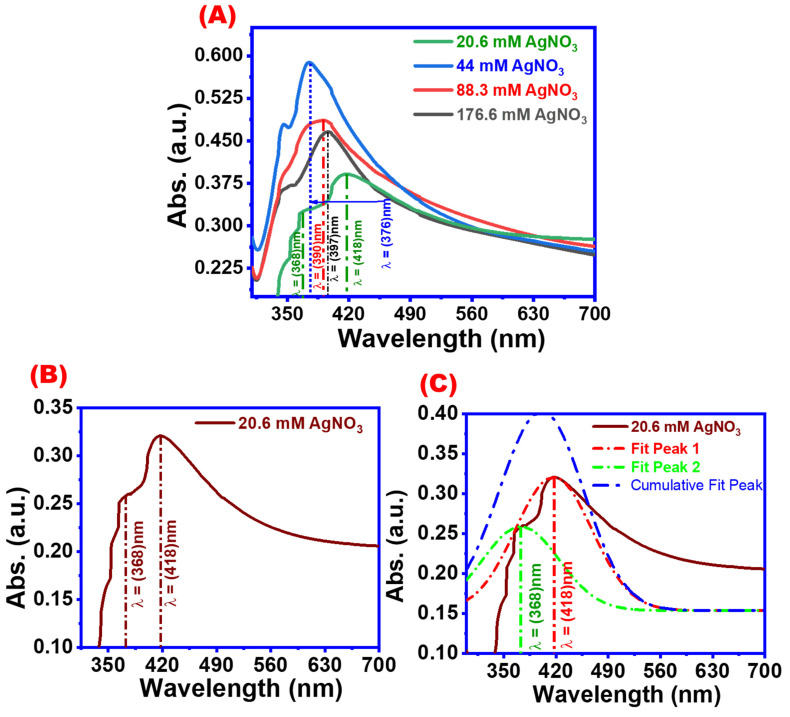
UV absorbance spectra of AgNWs (**A**) at different AgNO_3_ concentrations and (**B**,**C**) at 20.6 mM AgNO_3_ and the corresponding Gaussian fitting.

**Figure 8 materials-16-05501-f008:**
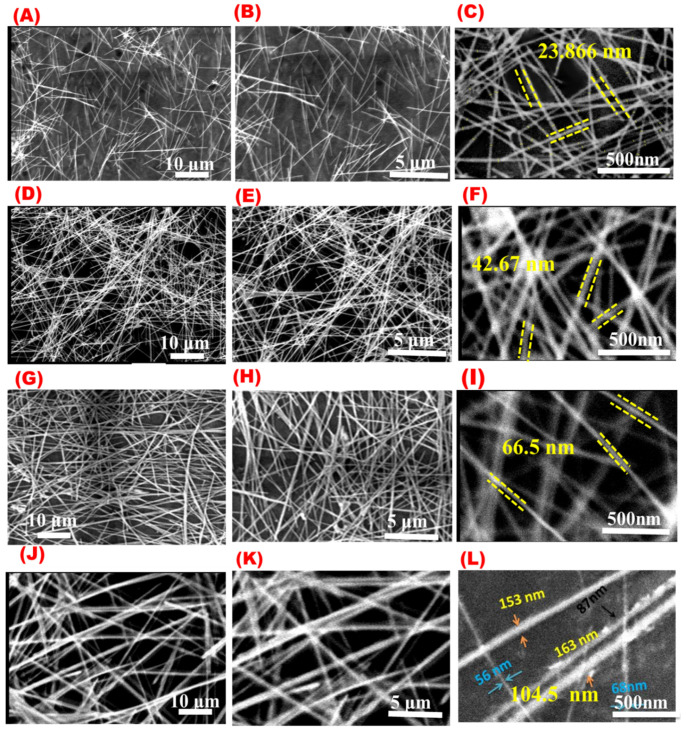
SEM images at different magnifications for AgNWs prepared at different agitation speeds: (**A**–**C**) at 700 rpm, (**D**–**F**) at 300 rpm, (**G**–**I**) at 100 rpm, and (**J**–**L**) without agitation.

**Figure 9 materials-16-05501-f009:**
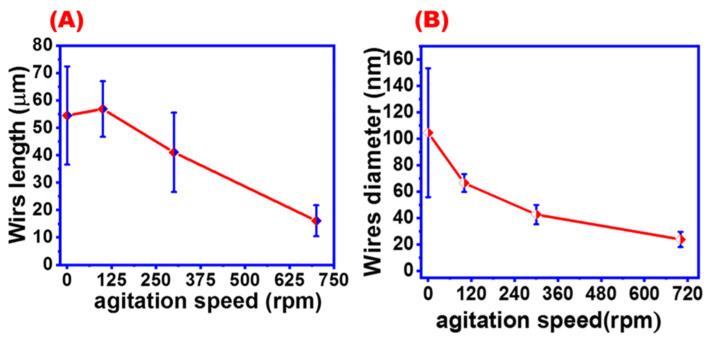
Effect of agitation speed on the (**A**) diameter of AgNWs and (**B**) length of AgNWs.

**Figure 10 materials-16-05501-f010:**
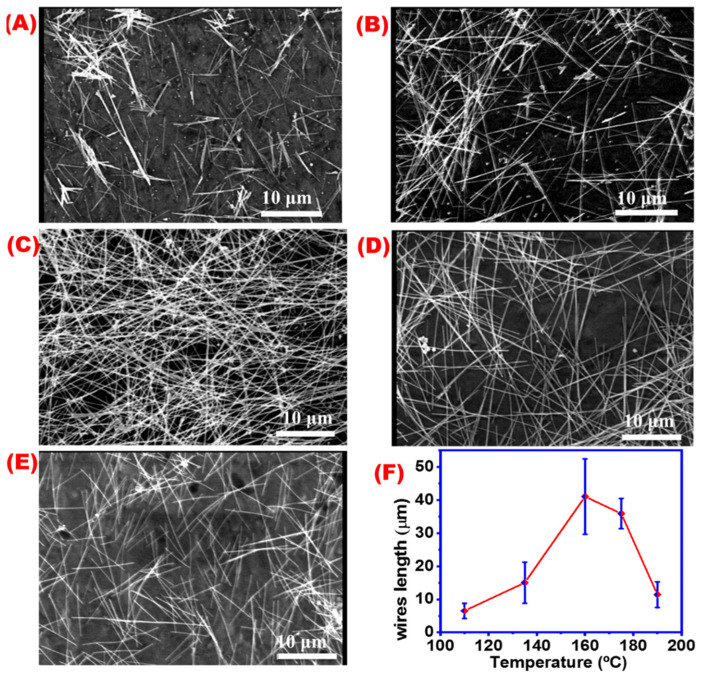
SEM images at 10 microns for AgNWs prepared under different temperatures: (**A**) 110 °C, (**B**) 135 °C, (**C**) 160 °C, (**D**) 175 °C, (**E**) 190 °C, and (**F**) effect of the reaction temperature on the length of the wires.

**Figure 11 materials-16-05501-f011:**
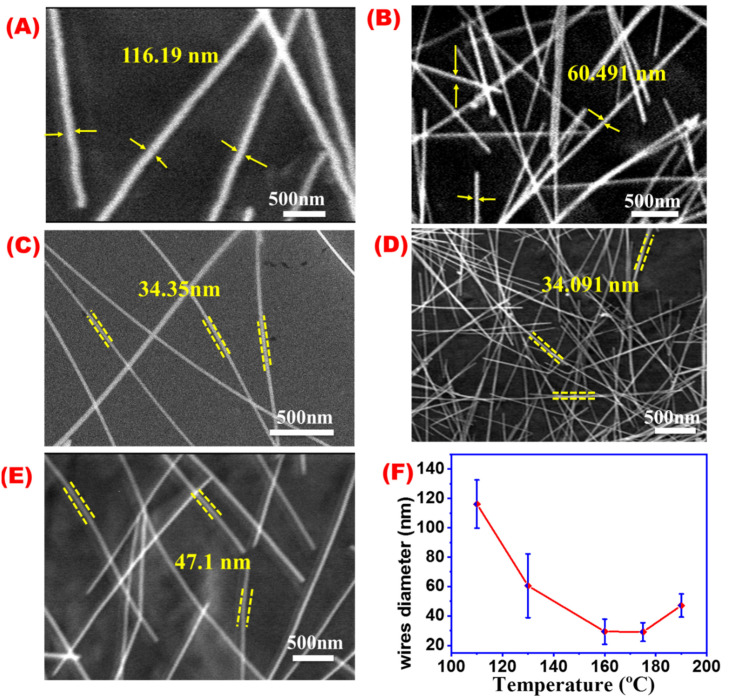
SEM images for AgNWs obtained at different temperatures; (**A**) 110 °C, (**B**) 135 °C, (**C**) 160 °C, (**D**) 175 °C, (**E**) 190 °C, and (**F**) effect of the reaction temperature on AgNW diameters.

**Figure 12 materials-16-05501-f012:**
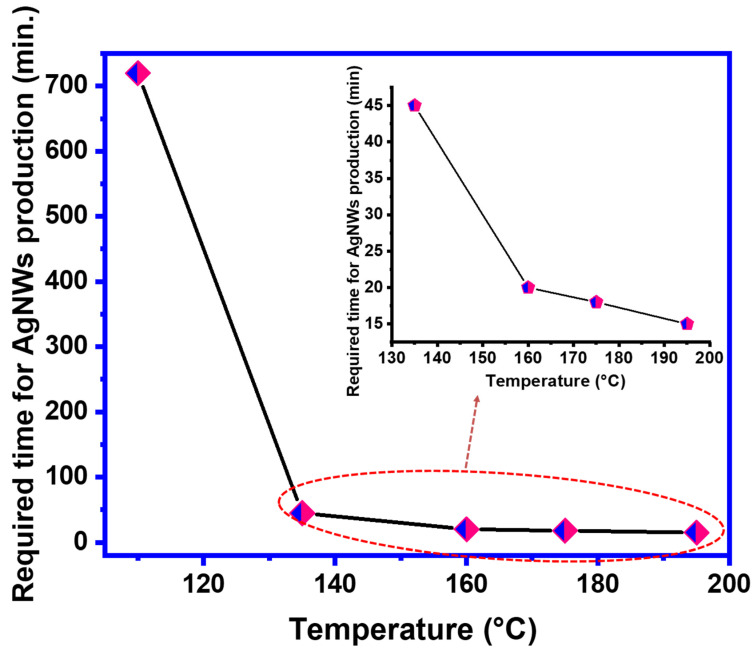
The effect of temperature on AgNW reaction time.

**Figure 13 materials-16-05501-f013:**
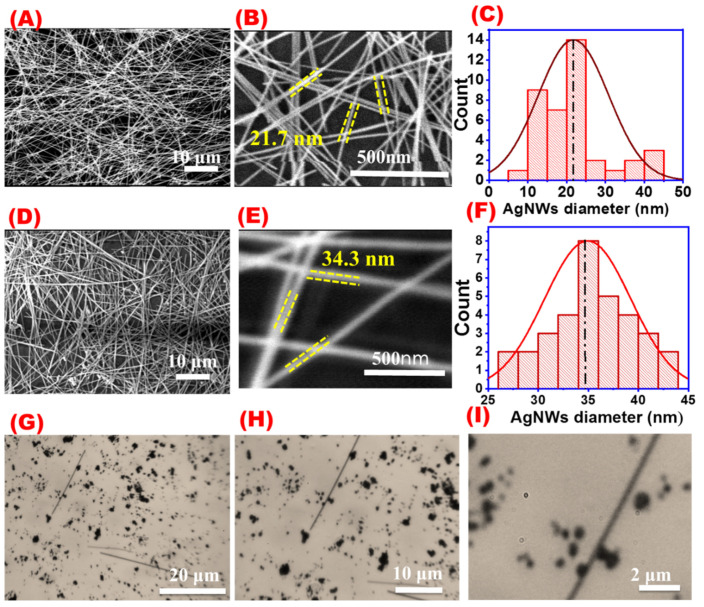
SEM images at different magnifications and the corresponding histogram for AgNWs obtained with different halides: (**A**–**C**) 2.6 mM KBr, (**D**–**F**) 2.6 mM KCl, and (**G**–**I**) 2.6 mM KI.

**Figure 14 materials-16-05501-f014:**
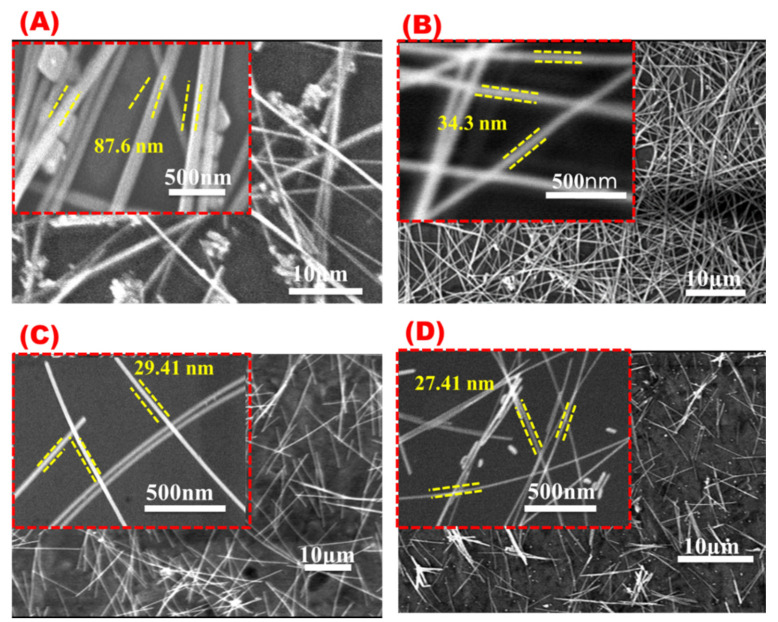
SEM images at different magnifications for AgNWs were obtained with different KCl concentrations: (**A**) 0.86 mM, (**B**) 2.6 mM, (**C**) 4.4 mM, and (**D**) 8.94 mM.

**Figure 15 materials-16-05501-f015:**
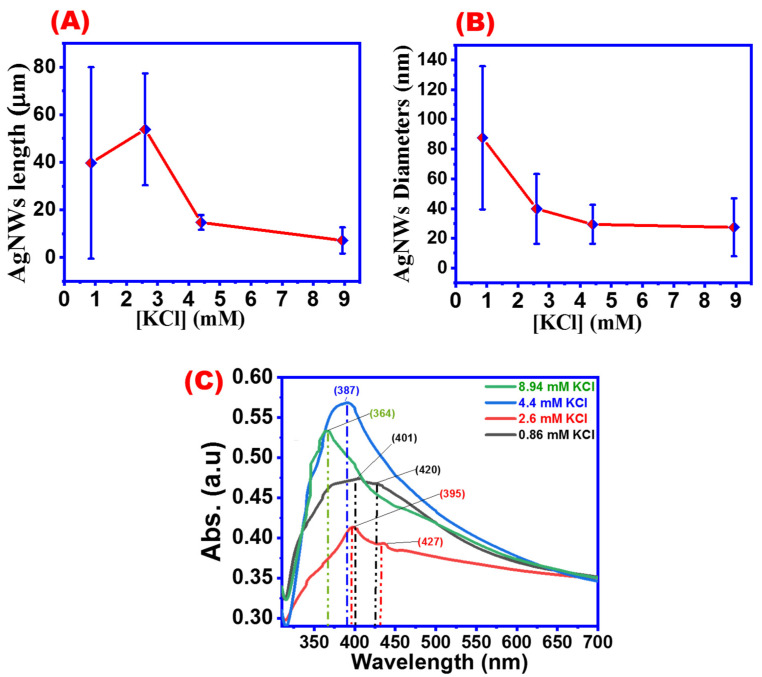
Effect of KCl concentrations on (**A**) AgNW diameter, (**B**) AgNW length, and (**C**) UV absorbance spectra of AgNWs obtained with different KCl concentrations.

**Figure 16 materials-16-05501-f016:**
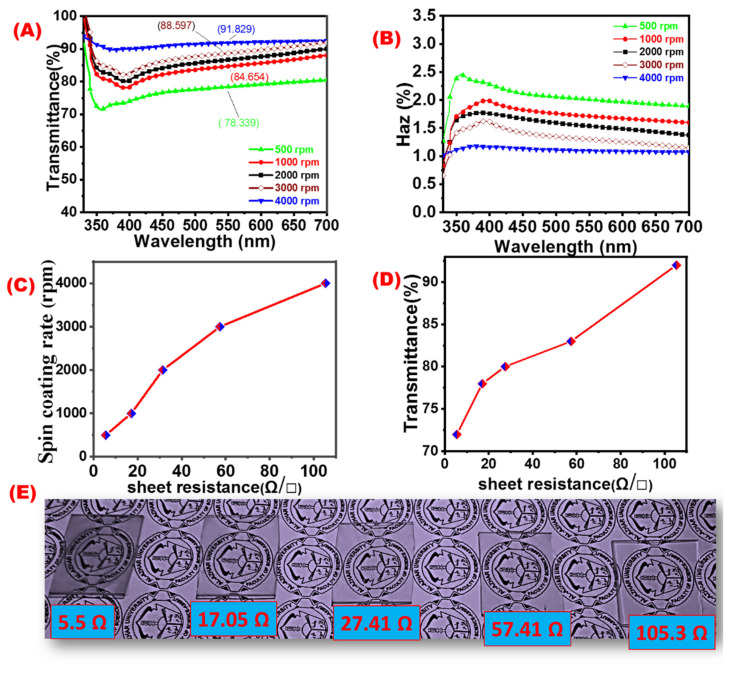
AgNWs-TCF designed under different spin coating rates: (**A**) transmittance (at λ = 550 nm), (**B**) optical haze, (**C**) sheet resistance, (**D**) transmittance, and (**E**) images of AgNWs-TCFs with different coating thickness.

**Figure 17 materials-16-05501-f017:**
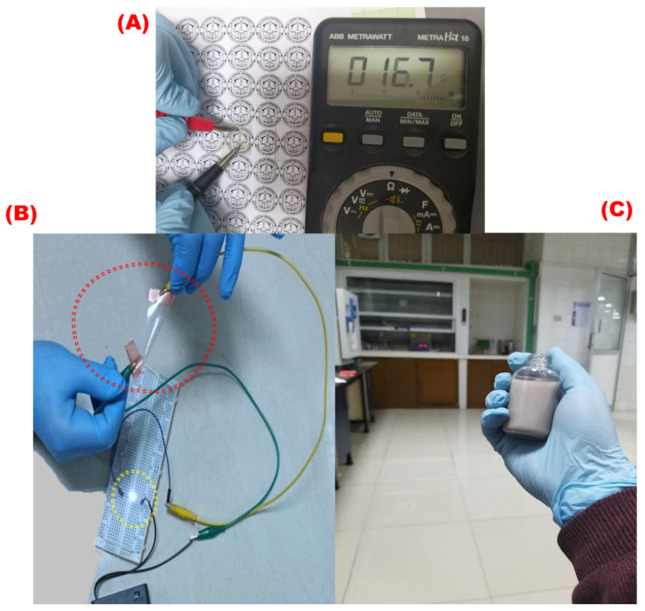
(**A**) AgNW electrode image with a sheet resistance of 17.3 “Ω/sq”, (**B**) electrical circuit for the flexible AgNW electrode, and (**C**) the AgNW ink with a concentration of 4 mg/mL.

## Data Availability

My manuscript and associated personal data.

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
