# Peer review of "Developing a Simple, Effective, and Quick Process to Make Silver Nanowires with a High Aspect Ratio"

_materials, 2023, doi:10.3390/ma16155501_

Round 1
Reviewer 1 Report
In this manuscript, the authors proposed a modified strategy for fabricating silver nanowires with a high aspect ratio efficiently, which is of significance for developing silver nanowires with outstanding optical and electrical properties in industry. Additionally, an electrode material with low electrical resistance and high transparency based on the silver nanowires was demonstrated in this work, suggesting a promising application in optical devices. However, the following concerns should be well addressed before the possible acceptance in Materials.
1. The unit such as millilitres and grams in line 76 should be written as mL and g, or should be presented in the same format, please correct.
2. There are several writing mistakes in figure captions. For example, letter number are not match with each figure in caption of Figure 5, and AgNO3 should be written as AgNO3 in the figure caption (line 235), and what is “Wavelengthλ” in Figure 7? please correct.
3. The units should only be written at the end of a series of numbers (such as line 241, 242, 243, 265, 358, 371 and 372), please check all text and correct.
4. There are also no spaces in many places in the results and discussion. For instance, there should be spaces for “0.005g” and “0.07M” in line 245 and 304, please check all text and correct.
4. There are also no spaces in many places in the results and discussion. For instance, there should be spaces for “0.005g” and “0.07M” in line 245 and 304, please check all text and correct.
5. In the article, there should be no space between the number and the temperature unit or the number and the percent (%) (such page 10 and 11), please check all text and correct.
6. The text of “Figure 6B” should be put into a parenthesis (line 213), and there are so many mistakes like that, please check all text and correct.
7. What is the writing format like “Figure 9. Summarizes” in line 250? please check all text and correct.
8. The first letter of the first word of a sentence should be capitalized, whereas there are several errors about it (line 259, 260, 290, 292 and 307), please check all text and correct.
9. The punctuation mark before however are lost, please correct.
10. There are a myriad of grammar mistakes (such as line 172, 199, 210, 222, 238, 250 and 264), please check all text and correct.
11. With the comparison of AgNWs prepared by traditional polyol method, how much improvement is there in electroconductibility?
12. The characteristics of the electrode prepared by AgNWs in this method such as stability, anti-erosion and some mechanical properties should be further investigated.
13. What is the difference between the electrode prepared by AgNWs in this method and the electrode prepared by AgNWs in the traditional method?
14. The comparative parameters such as performance, stability and cost between the electrode proposed in this work and the commercial electrode should be investigated and summarized.
15. The English of manuscript should be checked by native speaker.
Taken together, the manuscript should be subjected into the major revision before the possible acceptance in Materials.
see above listed items
Author Response
Dear Professor Editor of Materials
We would like to thank you and reviewers for yours and their valuable comments. Below we present a summary of the changes made to the manuscript according to raised comments and suggestions
Reviewer 1
Comments and Suggestions for Authors
In this manuscript, the authors proposed a modified strategy for fabricating silver nanowires with a high aspect ratio efficiently, which is of significance for developing silver nanowires with outstanding optical and electrical properties in industry. Additionally, an electrode material with low electrical resistance and high transparency based on the silver nanowires was demonstrated in this work, suggesting a promising application in optical devices. However, the following concerns should be well addressed before the possible acceptance in Materials.
- The unit such as millilitres and grams in line 76 should be written as mL and g, or should be presented in the same format, please correct this.
The response
The units had been corrected as follows
- 10 mL of ethylene glycol, 5 mL of KCl (0.005 g of KCl in 5 mL of ethylene glycol), 0.3 g of PVP solution (50% in ethylene glycol) and 0.15 g AgNO3 were mixed in a single container and heated to 160°C until the desired color was obtained.
- There are several writing mistakes in figure captions. For example, letter number are not match with each figure in caption of Figure 5, and AgNO3 should be written as AgNO3 in the figure caption (line 235), and what is “Wavelengthλ” in Figure 7? please correct this.
The response
Figure caption 5 had been completely corrected and λ had been deleted from the title on the X-axis in Fig 7.
Figure 5. SEM images at different magnification for AgNWs prepared with different AgNO3 concentration; (A-C) 20.6 mM of AgNO3, (D-F) 44 mM of AgNO3, (G-I) 88.3 mM of AgNO3 and (J-L) 176.6 mM of AgNO3
Figure 7. UV absorbance spectra of AgNWs; (A) at different AgNO3 concentrations and (B and C) at 20.6 mM AgNO3 and the corresponding Gaussian fitting.
- The units should only be written at the end of a series of numbers (such as line 241, 242, 243, 265, 358, 371 and 372), please check all text and correct.
The response
The correction had been done and applied to the manuscript for example:
- As was excepted, different agitation speeds will lead to different AgNWs morphologies. The average diameters of AgNWs obtained with agitation speeds of 700, 300, 100, and 0 rpm were found to be 866, 42.67, 66.5, and 104.5 nm, while the lengths of the wires were 16.08, 41.066, 56.86, 54.506 μm, respectively.
- There are also no spaces in many places in the results and discussion. For instance, there should be spaces for “0.005g” and “0.07M” in line 245 and 304, please check all text and correct.
The response
The spaces had been added for all text parts.
For example:
- Ultrathin AgNWs (20 nm) were obtained by using 005 g KBr which, acted as a co-nucleant that may inhibit the lateral growth of nanowires Figures 13 (A-C). An average length of 23.6 μm and an average diameter of 21.67 nm were obtained when using Br as a source of halogen in this study Figure 13C.
In the article, there should be no space between the number and the temperature unit or the number and the percent (%) (such page 10 and 11), please check all text and correct it.
The response
The spaces were deleted, and the correction had been applied to all text parts.
For example, .
- The light transmittance obtained for the corresponding samples were 23, 77.9, 81.3, 82.6 and 91.8% as shown in Figure 16A.
- The text of “Figure 6B” should be put into a parenthesis (line 213), and there are so many mistakes like that, please check all text and correct.
The response
The parenthesis had been added to all parts of the manuscript.
For examples:
- The average diameter was increased by increasing AgNO3 concentration, whereas the average AgNWs length reached a maximum at AgNO3 concentration of 44 mM (Figure 6B).
- A very thin wire with an average diameter of 25 nm and an average length of 8.49 μm also, a lot of nanoparticles have been produced with 20.6 mM AgNO3 (Figure 5(A-C)). Using a higher concentration of AgNO3 produces thicker AgNWs without AgNPs (Figure 5(D-L)).
- What is the writing format like “Figure 9. Summarizes” in line 250? please check all text and correct.
Text format had been corrected to times new roman, font 12 mm and line space of 1.5 mm.
Figure 9. Summarizes the dependence of AgNWs morphology on the agitation speed.
- The first letter of the first word of a sentence should be capitalized, whereas there are several errors about it (line 259, 260, 290, 292 and 307), please check all text and correct.
The response
The first letter of the first word of a sentence was capitalized and corrected for all text.
For examples:
- Figure Effect of agitation speed on the; (A) diameter of AgNWs and (B) length of AgNWs.
- The time required to produce AgNWs decreases from 12 hours to 20 minutes when the reaction temperature is raised from 110 to 160°C. These results are consistent with previous research [18, 45].
- The punctuation mark before however are lost, please correct them.
The response
- longer with chloride ions;[53] however, iodide is not valid for preparing AgNWs.
- Very thin AgNWs can be obtained by using a low concentration of AgNO3; however, silver nanoparticles will also be obtained as indicated by the presence of a UV peak at 418nm.
- There are a myriad of grammar mistakes (such as line 172, 199, 210, 222, 238, 250 and 264), please check all text and correct.
The response
We did, please see the revised version of the manuscript.
- With the comparison of AgNWs prepared by traditional polyol method, how much improvement is there in electroconductibility?
The response
The electroconductibility of silver nanowires was enhanced because the AgNWs prepared by the modified method are longer and thinner than those prepared by the traditional method. The transparent conductive electrodes depend on the ratio between the conductivity and the transparency. this ratio depends on the morphology of the prepared silver nanowires, and this was referred to in the manuscript. In addition, the advantages of the modern method over the traditional method have been added.
- The characteristics of the electrode prepared by AgNWs in this method such as stability, anti-erosion and some mechanical properties should be further investigated.
The response
We'll consider this valuable comments in the next paper
- What is the difference between the electrode prepared by AgNWs in this method and the electrode prepared by AgNWs in the traditional method?
The response
- The efficiency of the electrode depends on the morphological shape and purity of the synthesized AgNWs and this is extensively studied in the current work. Therefore, the difference between the electrode prepared with silver nanowires by the traditional method and the modern method is that:-
- The electrode designed by the modern method is very easy.
- Consumed a short time for production.
- The transparency of the electrode prepared by the traditional method is small and its resistance is very large compared to the electrode prepared with silver nanowires by the modern method.
- The comparative parameters such as performance, stability and cost between the electrode proposed in this work and the commercial electrode should be investigated and summarized.
The answered mentioned above
- The English of manuscript should be checked by native speaker.
The response
The whole manuscript has been revised and the English errors, grammatical and
typographical have been corrected
The manuscript has been resubmitted to your respected journal. I look forward to your positive response

Reviewer 2 Report
Journal: Materials
Manuscript ID: materials-2480344
Paper title: “Developing a simple, effective, and quick process to make silver nanowires with a high aspect ratio.”
Comments:
1. Title: Lacks technical accuracy and needs to be improved to better reflect the contents of the manuscript.
2. Method: Equation (1) needs to be shown clearly.
3. Results and Discussion: “Additionally, the somewhat greater ratio of 3.7 between the (111) and (200) planes compared to the theoretical ratio of 2.5 [27]” – Please elaborate this sentence especially on the ratio of 3.7.
4. Results and Discussion: How the average length of the AgNWs determined? It is not accurate since it depends on the size of the field of view of the SEM image captured every time before the analysis.
5. Results and Discussion: “Figure 6A shows the effect of AgNO3 concentration on the average diameter of 210 AgNWs. The average diameter was increased by increasing AgNO3 concentration, 211 whereas the average AgNWs length reached a maximum at AgNO3 concentration of 44 mM Figure 6B.” Why both trends are as mentioned above?
6. Conclusion: “We constructed different 410 AgNW electrodes that have low sheet resistance between 17.05 Ω.sq-1, and 105.3 Ω.sq-1, with transmittance greater than 76.8%, and 91.8% that can be introduced in optical devices and other applications.” Both cases need to be benchmarked against the commercial transparent electrodes in terms of sheet resistance and transmission.
7. References: Too many old references are used. The novelty of this work is therefore questionable.
Recommendation: Reject.
Average.
Author Response
I appreciate the valuable comments given by the reviewer that help us to improve the quality of the manuscript
Comments and Suggestions for Authors
The authors optimized the one-step, simple and reproducible modified polyol method to produce high-performance and high-throughput AgNWs. To obtain the best-optimized method, the parameters affecting the morphology of the silver nanowire have been significantly studied. They reached interesting findings, and the detailed results were illustrated in the manuscript. The manuscript is timely; however, I have some comments and questions from the authors and a re-review is mandatory:
- The contributions, applications, and especially, NOVELTIES of the paper should be clearly outlined in the last paragraph of the “Introduction” section to justify the motivation for this study. Shortcomings of the previous works should be highlighted. It is not very clear to a general reader.
The response
The required section had been added:
The previous strategies such as hard methods are very complicated, required a lot of conditions, and take a long time. Even the traditional polyol method requires more than 12 hours under inert gas, and it required more than one step [24]. Also, the morphology is not controlled, which motivates us to do this work.
- A deeper review on the research background should be provided in the "Introduction". No research for at least two years is observed in the References list. Therefore, the recent contributions should be discussed by the authors to enrich and update the literature review.
The response
The update has been added and our work is still distinguished compared to other researchers.
I compared our work with the following researcher according to the following references
[25] M. Parente, M. van Helvert, R. F. Hamans, R. Verbroekken, R. Sinha, A. Bieberle-Hütter, et al., "Simple and Fast High-Yield Synthesis of Silver Nanowires," Nano Letters, vol. 20, pp. 5759-5764, 2020/08/12 2020.
[26] G. Dzido, A. Smolska, and M. O. Farooq, "Rapid Synthesis of Silver Nanowires in the Polyol Process with Conventional and Microwave Heating," Applied Sciences, vol. 13, p. 4963, 2023.
[27] E. G. Yamamoto, M. P. Dantas, G. Yamanishi, F. B. Soares, A. Urbano, S. A. Lourenço, et al., "Silver nanowire synthesis analyzing NaCl, CuCl2, and NaBr as halide salt with additional thermal, acid, and solvent post-treatments for transparent and flexible electrode applications," Applied Nanoscience, vol. 12, pp. 205-213, 2022.
[28] K. Jhansi, N. Thomas, L. Neelakantan, and P. Swaminathan, "Controlling the aspect ratio of silver nanowires in the modified polyol process," Materials Letters, vol. 344, p. 134396, 2023.
- Please define every used abbreviation in its first usage place.
The response
The definition of all abbreviations is mentioned in its first usage place, and the text includes the definition of all abbreviations.
For example,
- Silver nanowires (AgNWs)
- AgNWs were characterized by Scanning electron microscope (SEM), X-ray diffraction (XRD), and Ultraviolet (UV) spectroscopy.
- Explain more about Fig. (7).
The response
The explanation had been done as shown below:
Figure7. Five characteristic UV absorbance peaks were observed at 368, 376, 390, 397 and 410 nm which are corresponding to AgNWs synthesized with 4 different AgNO3 concentrations. The first three peaks were attributed to AgNWs with average diameters of 270, 137 and 49.87 nm which were prepared under 176.6, 88.3 and 44 mM of AgNO3 concentrations, respectively. concerning with the two other peaks were characterized AgNWs with an average diameter of 25.43 nm and AgNPs, respectively. Based on these data, it is clear that with increasing the concentration of silver nitrate, the thickness of the wires increases, which leads to the emergence of UV absorption values towards a redshift (Figure 7A) [37]. Figures. 7 (B and C) show the UV absorbance spectra for AgNWs suspension sensitized with 20.6 mM AgNO3 and its corresponding Gaussian fitting, respectively. Two UV absorbance peaks were observed at 368 and 418 nm which are assigned to the thinner AgNWs, and AgNPs, respectively [38]. Very thin AgNWs can be obtained by using a low concentration of AgNO3; however, silver nanoparticles will also be obtained as indicated by the presence of a UV peak at 418 nm. It may be concluded that using a low concentration of silver source, not only will yield AgNWs with thinner diameter but also the suspension will contain silver nanoparticles. Consequently, to obtain uniform and thin wire with minimum amount of AgNPs, a concentration of 44 mM has been applied in this study.
- The “Results” section should be improved by adding explanations on the physical meaning of the graphs. The authors should justify each graph variation.
The response
We did, please see the revised version of the manuscipt
- There are some errors in English writing. So, it is recommended to double-check the whole manuscript to avoid grammatical and typo errors. Comments on the Quality of English Language
The response
The whole manuscript has been revised and the English errors, grammatical and
typographical have been corrected

Reviewer 3 Report
The authors optimized the one-step, simple and reproducible modified polyol method to produce high-performance and high throughput AgNWs. To obtain the best-optimized method, the parameters affecting the morphology of the silver nanowire have been significantly studied. They reached interesting findings, and the detailed results were illustrated in the manuscript. The manuscript is timely; however, I have some comments and questions from the authors and a re-review is mandatory:
1- The contributions, applications, and especially, NOVELTIES of the paper should be clearly outlined in the last paragraph of the “Introduction” section to justify the motivation for this study. Shortcomings of the previous works should be highlighted. It is not very clear to a general reader.
2- A deeper review on the research background should be provided in the "Introduction". No research for at least two years is observed in the References list. Therefore, the recent contributions should be discussed by the authors to enrich and update the literature review.
3- Please define every used abbreviation in its first usage place.
4- Explain more about Fig. (7).
5- The “Results” section should be improved by adding explanations on the physical meaning of the graphs. The authors should justify each graph variation.
6- There are some errors in English writing. So, it is recommended to double-check the whole manuscript to avoid grammatical and typo errors.
There are some errors in English writing. So, it is recommended to double-check the whole manuscript to avoid grammatical and typo errors.
Author Response
I appreciate the valuable comments given by the reviewer that help us to improve the quality of the manuscript
Comments and Suggestions for Authors
This study is so interesting for publishing after considering the following points.
- Can you please enhance the introduction section with more information about the importance of Ag nanowires for different applications?
The information about the importance of Ag nanowires for different applications had been added to the introduction part.
As a result of the unique optical and electrical properties of silver nanowires, there are many applications, based on AgNWs. Examples of applications based on silver nanowires are flexible displays, biosensors, electronic textiles, artificial organs, and other portable and wearable electronic devices. Also, applications of power supplies and transparent conductive electrodes. Most photovoltaic devices depend on transparent conductive electrodes to be suitable for some desired photovoltaic applications, so silver nanowires were the best in designing transparent electrodes[9-12].
- What is the main research question of this study?
How can you synthesize uniformed AgNWs in a few minutes with control of their length and diameter?
Some papers can help you relate to the plasmonic effects, such as https://doi.org/10.1016/j.ijleo.2018.07.135.
Very nice work and associated with our work.
[42] M. A. Basyooni, A. M. Ahmed, and M. Shaban, "Plasmonic hybridization between two metallic nanorods," Optik, vol. 172, pp. 1069-1078, 2018/11/01/ 2018.
- what are the potential ways to prepare aligned Ag nanowires or arrays.
The manuscript presented this clearly the factors affecting the morphology of silver nanowires, and these factors are summarized in 44 mM of AgNO3 and 0.005 g KCl and a temperature of 160 oC with an agitation speed of 100 rpm as the optimization parameters to produce uniformed AgNWs. Moreover, to obtained aligned Ag nanowires or arrays must be used halide 0.005 g Br- and a small concentration of 20.8 mM of AgNO3.
- in fig 15 (c), what is the role of the main peak of 387nm
The main peak at 387 nm may be due to the higher concentration, but it is not insignificant. It is important to change the location of the peak towards a higher or lower wavelength due to the change in the diameters of the prepared silver nanowires, which depend on the studied factor.
- in line 404, check the method of writing “160 c.”
It had been corrected.
reactants in one pot and using a small agitation of 100 rpm, a temperature of 160 ºC, 150 mM
- can you compare your results with the previously reported papers in preparation methods of Ag and highlighting the advantages and disadvantages
This part had been added to the introduction may be include some information about this comment.
The previous strategies such as hard methods are very complicated, required a lot of conditions, and take a long time. Even the traditional polyol method requires more than 12 hours under inert gas, and it required more than one step [28-30]. Also, the morphology is not controlled, which motivates us to do this work [29, 31, 32].
The advantages of our method for preparing silver nanowires can be summarized as follows:
- Simplicity, the other method more complicated [28-30]..
- Short time consumed with compared with other researcher [28-30].
- Prepared in one step [28-30].
- By the end of the manuscript, can you add an outlook section explaining the possibilities of preparing your methods for another metal such as Au, or,,,,,,, how will the difficulties
This part had been added to the conclusion part.
This method may be valid for the preparation of other metal nanowires such as Cu, Au and Al but, it requires the replacement of the capping agent with another type.
Comments on the Quality of English Language
Extensive editing of English language required
The response
The whole manuscript has been revised and the English errors, grammatical and
typographical have been corrected

Reviewer 4 Report
This study is so interesting for publishing after considering the following points.
- Can you please enhance the introduction section with more information about the importance of Ag nanowires for different applications?
- What is the main research question of this study?
Some papers can help you relate to the plasmonic effects, such as https://doi.org/10.1016/j.ijleo.2018.07.135
- what are the potential ways to prepare aligned Ag nanowires or arrays
- in fig 15 (c), what is the role of the main peak of 387nm
- in line 404, check the method of writing “160 c.”
- can you compare your results with the previously reported papers in preparation methods of Ag and highlighting the advantages and disadvantages
- By the end of the manuscript, can you add an outlook section explaining the possibilities of preparing your methods for another metal such as Au, or,,,,,,, how will the difficulties
Extensive editing of English language required
Author Response

(The authors gave the same response as above.)

Round 2
Reviewer 1 Report
All concerns have been addressed. It can be published as it now.
Author Response
Thank you very much for your kind opinion and accept our manuscript
Reviewer 2 Report
The comments were not addressed.
No issue.
Author Response
Dear respected Editor
The reviewers' suggestions have been taken into account. These suggestions will be incredibly helpful as we rewrite our work. We carefully considered the comments and updated the manuscript as a result. Below, we provide our responses in a point-by-point format.
Each response was emphasized in yellow and highlighted.
Additionally, we extensively edited the ENTIRE text to remove any grammar or syntax errors. Additionally, we have requested a number of colleagues who have experience writing academic papers in English to proofread the language. The phrasing is now, in our opinion, appropriate for the review procedure.
Below, we've highlighted the updated manuscript.
The Second Reviewer.
Comments:
- Title: Lacks technical accuracy and needs to be improved to better reflect the contents of the manuscript.
The response
We have chosen a title that reflects the contents of our paper, which focus on the optimization of our new method. However, we may suggest another title as follows:
One-Step Modified Polyol Method for Efficient Synthesis of High-Performance AgNWs
=================================================================================
- Method: Equation (1) needs to be shown clearly.
The response
To further illustrate the method of silver creation, additional equations have been added to equation 1 and discussed.
The missing equations have been added, and the text has been corrected as follows:
As shown in Equation (1), in the presence of halide anions as metal chlorides or metal bromides, AgNO3 reacts with chlorides or bromides to form AgX, which promotes the reduction process of Ag+ and can control silver concentration. [35, 36]. In the second step, Ag+ ions were reduced by EG to form silver atoms (seeds) as explained in Equations (2 and 3). The concentration of the seeds reaches the level of supersaturation at which the nucleation of Ag atoms takes place and they start to grow into silver nanostructures in the solution phase[33, 34].
=================================================================================
- Results and Discussion: “Additionally, the somewhat greater ratio of 3.7 between the (111) and (200) planes compared to the theoretical ratio of 2.5 [27]” – Please elaborate this sentence especially on the ratio of 3.7.
The response
The sentence had been explained, as follows:
Additionally, the intensity ratio between the (111) and (200) peaks is shown to be 3.5 compared to the theoretical value of 2.5, which may indicate the enhancement of the (111) crystal plane in AgNWs as mentioned in the previous studies [31, 32, 40, 41].
==================================================================
- Results and Discussion: How the average length of the AgNWs determined? It is not accurate since it depends on the size of the field of view of the SEM image captured every time before the analysis.
The response
Using ImageJ software, the average length of AgNWs was determined from several SEM images of AgNWs. The graphs also included statistical computations, such as mean value and standard deviation, which were clearly shown.
==================================================================
- Results and Discussion: “Figure 6A shows the effect of AgNO3 concentration on the average diameter of 210 AgNWs. The average diameter was increased by increasing AgNO3 concentration, 211 whereas the average AgNWs length reached a maximum at AgNO3 concentration of 44 mM Figure 6B.” Why both trends are as mentioned above?
The response
The manuscript presents the optimization of the synthesis of silver nanowires (AgNWs) with the desired morphology. The length of AgNWs can be increased by increasing the concentration of AgNO3, the silver nitrate precursor, up to 44 mM. However, at higher concentrations, the wires may become unstable and break apart. This is likely due to the increased concentration of silver ions, which can lead to the formation of larger crystals and/or the aggregation of AgNWs.
==================================================================
- Conclusion: “We constructed different 410 AgNW electrodes that have low sheet resistance between 17.05 Ω.sq-1, and 105.3 Ω.sq-1, with transmittance greater than 76.8%, and 91.8% that can be introduced in optical devices and other applications.” Both cases need to be benchmarked against the commercial transparent electrodes in terms of sheet resistance and transmission.
The response
The comparison had been added in the text:
The performance of AgNWs mesh transparent electrodes has been compared to indium tin oxide (ITO) and carbon nanotube (CNT) electrodes. It was found that AgNWs electrodes had comparable sheet resistance and optical transmittance to ITO and CNT electrodes [59]. Additionally, AgNWs electrodes showed superior flexibility and mechanical stability compared to ITO and CNT electrodes. Another study investigated the scalability of AgNWs transparent electrodes by studying their coating properties on large-area substrates. The performance of AgNWs electrodes was compared to ITO films in terms of sheet resistance, optical transmittance, flexibility, and reliability under bending conditions. The results showed that AgNWs electrodes could achieve similar or better performance than ITO films while being more scalable and cost-effective [60]. In addition, a comparison of AgNWs transparent electrodes with other works showed that the electrode resistivity designed from AgNWs was 27.41 Ω/sq, which was lower than the resistivity of 79Ω/sq at similar transparency for other works [61]. These results suggest that AgNWs transparent electrodes are an excellent choice for transparent electrodes.
======================================================================= electrodes.
- References: Too many old references are used. The novelty of this work is therefore questionable
The response
Updates have been made to the references as follows
[1] Z. Huang, J. Xu, Q. Zhang, G. Liu, T. Wu, T. Lin, et al., "Low-temperature polyol synthesis of millimeter-scale-length silver nanowires enabled by high concentration of Fe3+ for flexible transparent heaters," Materials Today Chemistry, vol. 30, p. 101569, 2023
.
[2] L. Yang, X. Huang, H. Wu, Y. Liang, M. Ye, W. Liu, et al., "Silver Nanowires: From Synthesis, Growth Mechanism, Device Fabrications to Prospective Engineered Applications," Engineered Science, vol. 23, p. 808, 2023.
[6] M. Bian, Y. Qian, H. Cao, T. Huang, Z. Ren, X. Dai, et al., "Chemically Welding Silver Nanowires toward Transferable and Flexible Transparent Electrodes in Heaters and Double-Sided Perovskite Solar Cells," ACS Applied Materials & Interfaces, vol. 15, pp. 13307-13318, 2023.
[7] S. Devaraju, A. K. Mohanty, D.-h. Won, and H.-j. Paik, "One-step fabrication of highly stable, durable, adhesion enhanced, flexible, transparent conducting films based on silver nanowires and neutralized PEDOT: PSS," Materials Advances, vol. 4, pp. 1769-1776, 2023.
[8] I. Ibrahim Zamkoye, J. Bouclé, N. Leclerc, B. Lucas, and S. Vedraine, "Silver Nanowire Electrodes Integrated in Organic Solar Cells with Thick Active Layer Based on a Low‐Cost Donor Polymer," Solar RRL, vol. 7, p. 2200756, 2023.
[22] Y. Li, Y. Wang, J. Wu, Y. Pan, H. Ye, and X. Zeng, "Synthesis of Silver Nanowires Using a Polyvinylpyrrolidone-Free Method with an Alpinia zerumbet Leaf Based on the Oriented Attachment Mechanism," ACS omega, vol. 8, pp. 2237-2242, 2023.
[23] N. Nasikhudin, Y. Al Fath, H. Rahmadani, M. Diantoro, H. Pujiarti, and S. Abd Aziz, "Propylene Glycol and Glycerol Addition in Forming Silver Nanowires (AgNWs) for Flexible and Conductive Electrode," in E3S Web of Conferences, 2023, p. 01020.
[58] Z. Fan, J. Wang, L. He, B. Shen, J. Chen, H. Mao, et al., "Tetrabutylammonium Tribromide-Induced Synthesis of Silver Nanowires with Ultrahigh Aspect Ratio for a Flexible Transparent Film," Langmuir, 2023.
[61] M. Saeidi, A. Eshaghi, and A. A. Aghaei, "Electro-optical properties of silver nanowire thin film," Journal of Materials Science: Materials in Electronics, vol. 34, p. 110, 2023.
The third reviewer
Comments on the Quality of English Language
It seems the authors didn't respond the 1st round review comments made by this reviewer. Please consider them.
The response
We have revised the WHOLE manuscript carefully and tried to avoid any grammar or syntax error. In addition, we have asked several colleagues who are skilled authors of English language papers to check the English. We believe that the language is now acceptable for the review process.
The fourth reviewer
Comments on the Quality of English Language
Extensive editing of English language required
The response
We have revised the WHOLE manuscript carefully and tried to avoid any grammar or syntax error. In addition, we have asked several colleagues who are skilled authors of English language papers to check the English. We believe that the language is now acceptable for the review process.
The manuscript has been resubmitted to your respected journal. I look forward to your positive response.
Again we thank the reviewers' and editors.
We look forward to your kind response.
Sincerely Yours,
- R. Shaaban
Corresponding author

Reviewer 3 Report
It seems the authors didn't respond the 1st round review comments made by this reviewer. Please consider them.
It seems the authors didn't respond the 1st round review comments made by this reviewer. Please consider them.
Author Response
We have revised the WHOLE manuscript carefully and tried to avoid any grammar or syntax error. In addition, we have asked several colleagues who are skilled authors of English language papers to check the English. We believe that the language is now acceptable for the review process.
Please, see the revised version of the manuscript.
Reviewer 4 Report
thank you for your comments
Extensive editing of English language required
Author Response

(The authors gave the same response as above.)
